# OpenReview forum: "MultiVerse: Exposing Large Language Model Alignment Problems in Diverse Worlds"
_NeurIPS.cc/2024/Workshop/SafeGenAi — SafeGenAi Poster_

### Official Review · Reviewer_DtMy · 2024-10-08
**A good jailbreak paper in simulated "worlds"**

**Rating:** 6
**Confidence:** 3

**Review:**

This paper introduces a method called MULTIVERSE to highlight and study alignment vulnerabilities in Large Language Models (LLMs). The authors present a Domain-Specific Language (DSL) that describes various possible contexts, called "worlds," allowing systematic generation of diverse conversation settings. By embedding malicious questions within these worlds, the paper demonstrates how LLMs can be deceived into bypassing safety mechanisms, revealing latent alignment issues. Through large-scale experiments, the authors show that LLMs are particularly vulnerable in contexts far removed from real-world scenarios, especially in nested and programming-related worlds. The paper claims that current LLM alignment training is insufficient when addressing virtual contexts, raising significant security concerns.

---
## Strength
- **Innovative Approach:** The paper introduces a novel way to expose LLM vulnerabilities by systematically generating diverse virtual worlds using DSL. This method provides a fresh perspective on alignment issues.
- **Comprehensive Evaluation:** Extensive experiments are conducted on both open-source and closed-source LLMs, offering strong empirical evidence for the effectiveness of the MULTIVERSE approach.
- **Scalable Methodology:** The approach is cost-effective and can scale to a large number of generated worlds, making it practical for identifying LLM vulnerabilities in various contexts.

---
## Weakness
- **Limited Focus on Defense Mechanisms:** While the paper highlights significant vulnerabilities in LLM alignment, it does not explore in-depth how to mitigate these issues or propose robust defense mechanisms.

---
## Suggested Improvement
- **Add a discussion section for the defense approach of this jailbreak method.**
- **Split the appendix into various sections.** Do not merge them all together in one section for clarity of your paper.
- **Typos and writing:** quote and unquote problems. For example, in Line 239, the authors use a mixture of " and '. I think the writing of this paper should be improved.

---

### Official Review · Reviewer_h7Qy · 2024-10-09
**Excellent systematic method for generating jailbreaks using nested "worlds"**

**Rating:** 8
**Confidence:** 4

**Review:**

quality
-------
Pros
- 300 different worlds is a good sample size, and 2 popular LLMs is sufficient to show generalisation.
- Good number of comparisons to baselines
- The Domain-specific language is an interesting addition, at the risk of being over-kill but in practice helps unambiguously notate and clarify the prompts' structures. This advantage is noted ~L118: "Note that although LLMs are capable of generating free-form universes, our goal is to systematically study the contexts in which LLMs are vulnerable, and hence using WDL is a better design choice." In particular, it allows easy extension to automated multi-round jailbreak (§4.3).
~L218: Good to have looked into the JSR drop at 5-layers.
- Good to investigate the sensitivity to various parameters e.g. location, time, etc

Cons
- ~L177 You mention Chinese language: how many of the nested worlds rely on using non-English languages?
- You claim that the method is cost-effective, but don't carry it out on GPT-4 for cost reasons.


clarity
-------
Pros
- The introduction is a very smooth introduction into the paper
- Graphs and figures are very clear

Cons
- The introduction is slow to start, we could perhaps assuming knowledge of RLHF and jailbreaking and jump straight into Figure 1.
- Not clear to me why some links are footnotes instead of included with the rest of the bibliography
- ~L170: you say "i.e. ChatGLM..." and "e.g. Gemini...." - are these the actual models used, or simply illustrative examples? If the latter, strike "ie" and "eg"
~L231 / Observation 3: This seems poorly phrased, I don't really understand what's being said, though I can infer.



originality
---------
Pros
- While automated jailbreaking is an established field, this is (to my knowledge) the first example of "nesting worlds", certainly in such a formal and automatable form.

significance
------------
Pros
- The method of nesting "worlds" as a general-purpose way to jailbreak models is highly significant, and the paper establishes a technique to automatically create such prompts with high success rate.
-  Another key finding: "updating parameters such as Location and Language to formulate a more complex context combination enhances the jailbreak success rate"


misc other
----------
the link [2] to jailbreakchat.com is broken

---

### Official Review · Reviewer_uUjD · 2024-10-09
**Multiverse paper: requires more clalrificaitons**

**Rating:** 6
**Confidence:** 3

**Review:**

Strength:
1. The authors proposed using a structured format (DSL) to automate the generation of jailbreaking prompts using words from WDL dataset.
2. The authors provided detailed evaluations to quantify their results with different models. using metrics like JSR, AQQ .
3. The sentivity analysis, including the number of layers, etc. , demonstrated that LLMs are vulnerable to jaibreaking in scneariso that are complex and nested.

Weakness:
1. Based on my understanding, the proposed method is actually a semi-automatic jailbreaking method based on pre-selected prompt architecture and words (DSL) created using LLMs. The part that generates the jailbreaking prompt is indeed automatic.
In line 567, under the section Jailbreak LLMs, the authors state that the proposed approach "efficiently ...generate jailbreak prompts inside a nesting of multiple fantasy worlds by MULTIVERSE. "
My question is that if we use prompt enginering by passing your prompt structure in Figure 3, 4, or even samples provided later, the automatic LLM-based jalbreaking algorithms can potentially achieve jailbreaking only using "a nesting of multiple fantasy worlds by MULTIVERSE."  For example, by passing this template to AutoDAN, this may potentially generate better results.

2. Following on the previous comment, the claim of low cost on line 10 of the proposed needs further clarification:
By switching to a different model/dataset/domain, etc., the cost associated with DSL, complier, would the proposed method be more or less expensive when compared to automatic LLM-based jailbreaking methods? Is cost associated with initial work taken into account?
Please elabroate.